# Resonating holes vs molecular spin-orbit coupled states in group-5 lacunar spinels

Thorben Petersen [1] ✉, Pritam Bhattacharyya[1], Ulrich K. Rößler [1] &
Liviu Hozoi[1] ✉

The valence electronic structure of magnetic centers is one of the factors that determines the characteristics of a magnet. This may refer to orbital degeneracy, as for $j_{eff} = 1/2$ Kitaev magnets, or near-degeneracy, e.g., involving the third and fourth shells in cuprate superconductors. Here we explore the inner structure of magnetic moments in group-5 lacunar spinels, fascinating materials featuring multisite magnetic units in the form of tetrahedral tetramers. Our quantum chemical analysis reveals a very colorful landscape, much richer than the single-electron, single-configuration description applied so far to all group-5 Ga$M_4$$X_8$ chalcogenides, and clarifies the basic multiorbital correlations on $M_4$ tetrahedral clusters: while for V strong correlations yield a wave-function that can be well described in terms of four $V^{4+}V^{3+}V^{3+}V^{3+}$ resonant valence structures, for Nb and Ta a picture of dressed molecular-orbital $j_{eff} = 3/2$ entities is more appropriate. These internal degrees of freedom likely shape vibronic couplings, phase transitions, and the magneto-electric properties in each of these systems.

A magnet is a collection of magnetic moments; its characteristic properties are determined by the nature of those moments and by how they mutually interact. To shape the properties of magnetic materials according to specific requirements we therefore need to (i) understand and (ii) have some degree of control over magnetic moments—inner structure and the way they interact with each other. It turns out that both—inner morphology and mutual interaction—depend on the subtle interplay of electronic correlations, spin-orbit couplings (SOCs), and crystal-field effects (CFEs). The combined action of these three factors received enormous attention in recent years. New insights and new ideas in this research area have led to new physical models, new concepts, and new research paths, as for example Kitaev's spin model[1] and extensive associated work[2].

Here we reveal what lies behind effective moments in each of the group-5 Ga$M_4$$X_8$ lacunar spinels ($M$ = V, Nb, Ta and $X$ = S, Se), fascinating materials displaying remarkable magnetic[3], magneto-electric[4,5], and transport[6,7] properties. The characteristic structural feature of this family of compounds is that the transition-metal ions are clustered as $M_4$ tetrahedra (see Fig. 1). The latter can be then viewed as effective

(magnetic) sites of a fcc lattice and their electronic structure can be described in terms of $T_d$ point-group symmetry-adapted cluster orbitals—$a_{1/2}$, $e$, and $t_{1/2}$. From electronic-structure calculations based on density functional theory (DFT), an $a_1^2 e^4 t_2^1$ valence electron configuration was inferred[8–10]. While indications of genuine many-body physics are available from both ab initio quantum chemical investigations[11–13] and dynamical mean field theory (DMFT)[14], an in-depth profile of correlation effects across the 3d-4d-5d lacunar-spinel series is missing, which is the scope of our present quantum chemical study.

Besides clarifying essential electronic-structure features, the multiconfigurational wave-function analysis that we provide—in terms of either localized, site-centered or multisite orbitals—makes these materials a distinct correlated-electron model system, as illustrative but much more captivating than other platforms typically employed to illustrate electronic correlations, as, e.g., the $H_2$ molecule for variable interatomic separation[15,16]. Using as indicator for the strength of correlations the weight of ionic configurations in the ground-state wave-function, we picture (i) what strong correlations mean in the 3d

[1]Institute for Theoretical Solid State Physics, Leibniz IFW Dresden, Helmholtzstraße 20, Dresden D-01069, Germany. ✉e-mail: t.petersen@ifw-dresden.de; l.hozoi@ifw-dresden.de

vanadates ($GaV_4S_8$, $GaV_4Se_8$, $AlV_4S_8$) and (ii) the notion of moderate correlations and 'dressed' $j_{eff} = 3/2$ objects in the $4d$ ($GaNb_4S_8$, $GaNb_4Se_8$) and $5d$ ($GaTa_4Se_8$) variants. Even for the heavier cations, when expressing the multiconfigurational wave-functions in terms of delocalized multisite orbitals, the weight of the $(\ldots)t_2^1$ configuration presently assumed to correctly describe the ground-state amounts to only 60%. Impressively, that shrinks to as little as 20% for $3d$ electrons. Yet, SOCs are still effective—even in the vanadates, those give rise to a spin-orbit-induced splitting of ≈10 meV for the ground-state term. Also peculiar here is the near-degeneracy of the ground and a higher-spin state, which should be possible to evidence by either spectroscopy or pressure experiments. All these electronic-structure features for the $3d$ case—massive correlations, scaled down but still detectable spin-orbit fine structure, and close proximity of high-spin states—outline a few important differences between $3d$ and $4d/5d$ group-5 lacunar spinels, i.e., the starting point in understanding the differences in their magnetic properties.

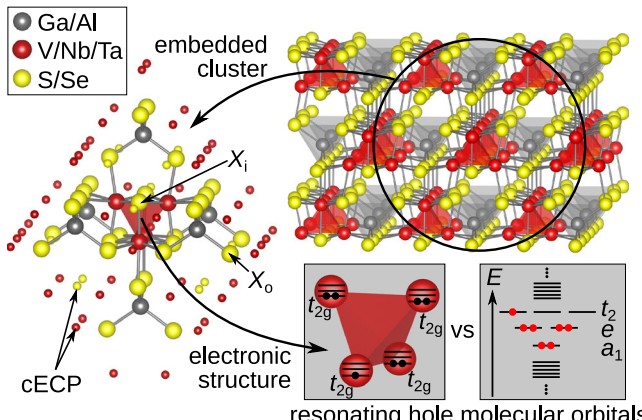

**Fig. 1 | Lacunar-spinel crystal structure and embedded cluster model employed in this study.** From the extended solid, a $[M_4X_{28}Ga_6]^{25-}$ cluster is cut for quantum chemical analysis ($M$ = V, Nb, Ta, and $X$ = S, Se). Each transition-metal ion $M$ is surrounded by six ligand ions $X$ in a distorted octahedron. $X_i$ and $X_o$ labels indicate $X$ atoms inside and outside the $M_4$ tetrahedron, respectively, referring to the basis set assignment in Supplementary Table 2. The small spheres indicate embedding capped effective core potentials (cECPs). Inset: valence level representations analyzed in this study.

## Results

### High-temperature, tetrahedral-symmetry multiplet structure

The $M_4$-tetrahedron multiplet structure, as computed for the high-temperature $F\bar{4}3m$ cubic lattice arrangement of group-5 lacunar spinels, is displayed in Fig. 2. The multiconfigurational complete active space self-consistent field (CASSCF) method[17] with an active space of seven electrons in twelve orbitals [(7e,12o)-CAS] was applied. Those orbitals are depicted in Supplementary Figure 1.

For $GaV_4S_8$ (Fig. 2a), this (7e,12o)-CASSCF yields a high-spin ($S$ = 3/2) ground state. Accounting for dynamical correlation effects in the scheme of *post*-CASSCF $N$-electron valence perturbation theory (NEVPT2)[18] corrects this state ordering. Near degeneracy of low- and high-spin states is an effect often seen in $3d$ systems, due to the similar magnitude of Coulomb interactions and various valence level splittings; in solid-state context, a well-known example is $LaCoO_3$ (see[19] for a quantum chemical investigation). Spin-crossover effects were also observed in DMFT calculations on $GaV_4S_8$[14]. In contrast, for the Nb- (Fig. 2b) and Ta-based materials (Fig. 2c), the CASSCF(7e,12o) methodology already ensures a reasonably good description—the NEVPT2 scheme provides only minor corrections to the relative energies in the $4d$ and $5d$ systems (see Supplementary Tables 4 and 5).

While a $^2T_2$ state is found as ground state in all compounds, the $a_1^2e^4t_2^1$ electronic configuration (where $a_1$, $e$, and $t_1$ are symmetry-adapted, molecular-like orbitals in $T_d$ point-group symmetry) contributes with weights well below 100% to the ground-state wave-functions; this aspect is discussed in detail in the following section. SOC further splits the degenerate $^2T_2$ components into a $j_{eff} = 3/2$ ground and a $j_{eff} = 1/2$ excited state in all instances, with the magnitude of this splitting increasing from 12 meV ($3d$) to 97 meV ($4d$) and 345 meV ($5d$ ions); the latter number, in particular, suggests that the origin of the peak found at ≈0.3 eV in resonant inelastic X-ray scattering (RIXS) measurements on $GaTa_4Se_8$[20] is spin-orbit splitting within the $^2T_2$ ($a_1^2e^4t_2^1$) levels, different from the initial interpretation in terms of $e^4t_2^1 \rightarrow e^3t_2^2$ transitions[12,20].

It is seen that the separation between the ground $^2T$ and the first excited $^4T$ state also increases for heavier transition-metal species: 41 meV in $GaV_4S_8$, 475 meV in $GaNb_4Se_8$, and 598 meV in $GaTa_4Se_8$. A denser set of low-lying excited-state levels in the vanadate could explain the significant deviation from Curie-Weiss behavior seen at higher temperatures in susceptibility measurements[21,22], although a quantitative analysis would require the inclusion of vibronic couplings and intersite magnetic interactions (see discussion in Section V of the Supplementary Information and in our previous work[12]). The latter then dictates the structure of the inelastic neutron scattering spectra[23].

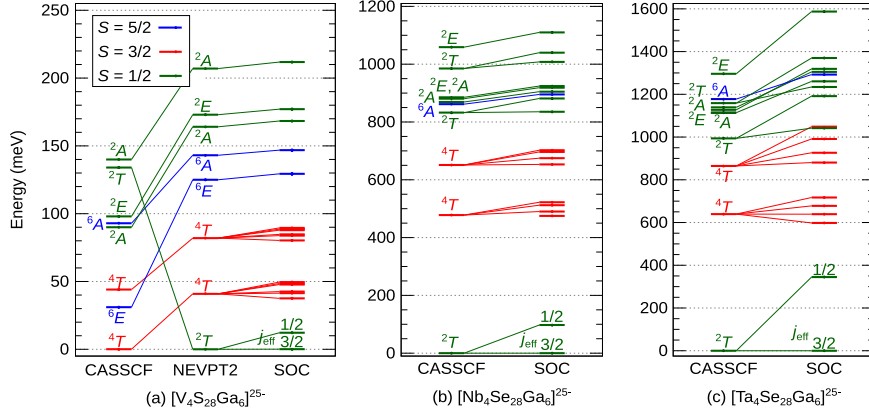

**Fig. 2 | Low-energy multiplet structures for $GaV_4S_8$, $GaNb_4Se_8$, and $GaTa_4Se_8$.** The excitation energies were calculated for (**a**) $[V_4S_{28}Ga_6]^{25-}$, (**b**) $[Nb_4Se_{28}Ga_6]^{25-}$, and (**c**) $[Ta_4Se_{28}Ga_6]^{25-}$ embedded clusters (CAS(7e,12o)). Different spin states $S$ ($S$ = 1/2, 3/2, 5/2) are shown in green, red, and blue, respectively. SOC was accounted for on top of the CASSCF/NEVPT2 wave functions. The corrections brought by NEVPT2 are minor for the Nb- and Ta-based compounds and therefore not depicted (see Supplementary Tables 4 and 5).

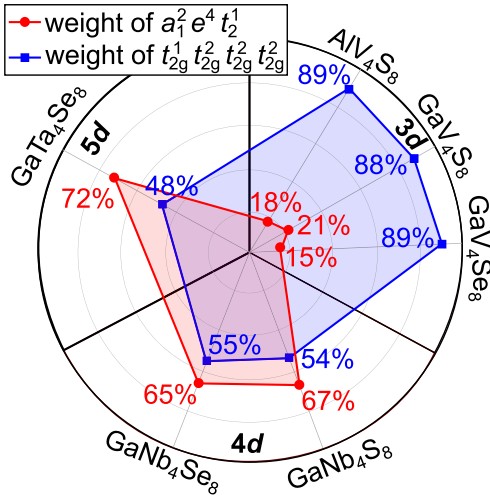

**Fig. 3 | Polar plot with weights for the leading configuration in symmetry-adapted (red) and localized (blue) orbital basis.** For each system, the analysis is performed in terms of ground-state optimized CASSCF(7e,12o) wave-functions.

### Ground-state correlations in cubic group-5 spinels

A major difference between how the single-tetrahedron electronic structure is presently depicted in the literature and in our quantum chemical results is the composition of the ground-state $^2T_2$ term. Different from the 100% $a_1^2 e^4 t_2^1$ ground state assumed so far on the basis of DFT computations for these materials, we find weights of 72% in GaTa$_4$Se$_8$, 65% in GaNb$_4$Se$_8$, and as little as 21% in GaV$_4$S$_8$ for the $a_1^2 e^4 t_2^1$ configuration. Other configurations contributing to the ground-state wave functions imply, for example, double excitations into higher-lying $t_1$ and $t_2$ levels, each of those configurations with a weight of a few percent or even less (see also Supplementary Table 6).

The much more pronounced multiconfigurational character of the vanadate ground-state wave function in the symmetry-adapted orbital basis is also reflected in the correlation index proposed by Ramos-Cordoba et al.[15], which serves as a measure for the extent of near-degeneracy effects (also referred to as nondynamical correlation in quantum chemistry). Using the (7e,12o) CASSCF natural-orbital occupation numbers, we derived nondynamical correlation indices $I_{\text{ND}}$[15] ranging from ≈2 in vanadates (1.96 for GaV$_4$S$_8$ and 2.00 for GaV$_4$Se$_8$) to ≈1.05 in the 4$d$ variants (1.05 for GaNb$_4$S$_8$ and 1.07 for GaNb$_4$Se$_8$) and 0.92 in GaTa$_4$Se$_8$ (details are given in the Supplementary Table 7). To put this in perspective, along the H$_2$ dissociation curve, $I_{\text{ND}}$ evolves from less than 0.1 at equilibrium distance to 0.5 towards dissociation[15], although a direct comparison of $I_{\text{ND}}$ values between chemically different systems is not straightforward.

Looking for further insight, we re-expressed the many-body ground-state wave functions in terms of atomic-like, single-site orbitals. The orbital localization module available in ORCA was employed for this purpose; from the 12 symmetry-adapted orbitals ($1 \times a_1$, $1 \times e$, $1 \times t_1$, $2 \times t_2$) obtained in the (7e,12o) CASSCF we arrive then to 12 $t_{2g}$-like functions (three per pseudo-octahedrally coordinated transition-metal ion, see Supplementary Figure 1). For both orbital bases, symmetry-adapted or site-centered, the weights of the leading configurations are illustrated in Fig. 3, for GaV$_4$S$_8$, GaV$_4$Se$_8$, GaNb$_4$S$_8$, GaNb$_4$Se$_8$, GaTa$_4$Se$_8$, and an $A$-site-substituted member of the family (see also Supplementary Table 6). Interestingly, a low weight of the $a_1^2 e^4 t_2^1$ configuration in the symmetry-adapted orbital basis is associated with large weight of the $t_{2g}^1 t_{2g}^2 t_{2g}^2 t_{2g}^2$ (i.e., V$^{4+}$V$^{3+}$V$^{3+}$V$^{3+}$) configurations in the localized-orbital representation; for the AlV$_4$S$_8$, GaV$_4$S$_8$, and GaV$_4$Se$_8$ vanadates, in particular, 15–20% $a_1^2 e^4 t_2^1$ translates into 85–90% configurations of $t_{2g}^1 t_{2g}^2 t_{2g}^2 t_{2g}^2$ type. The remaining part stems mainly from triply-occupied transition-metal centers, i.e., excited-state

configurations of $t_{2g}^1 t_{2g}^2 t_{2g}^1 t_{2g}^3$ (V$^{4+}$V$^{3+}$V$^{4+}$V$^{2+}$) type. Weights of only ≈9% for the latter indicate much stronger correlations in the case of V-based lacunar spinels: intersite fluctuations are heavily suppressed, compared to the 4$d$ and 5$d$ compounds. In a Mott-Hubbard picture, stronger localization is the result of larger $U/t$ ratio. In other words, correlations are moderate in the 4$d$ and 5$d$ compounds and strong in the vanadates—for the latter, an expansion in terms of four V$^{4+}$V$^{3+}$V$^{3+}$V$^{3+}$ resonant valence structures already provides a reasonably good description. For perspectives on valence bond theory, the reader is referred to e.g.[24,25].

### Discussion

To analyze in detail how correlations evolve from 3$d$ to 4$d$ and 5$d$ ions for the same type of leading ground-state configuration and in the same crystallographic setting is difficult. 3$d^5$ (Mn$^{2+}$, Fe$^{3+}$) species, for example, tend to adopt a $t_{2g}^3 e_g^2$ ground-state electron configuration, while 4$d^5$ (Ru$^{3+}$, Rh$^{4+}$) and 5$d^5$ (Ir$^{4+}$) varieties display a $t_{2g}^5$ valence-orbital occupation. Thinking of lower $d$-shell filling, Mo$^{5+}$ 4$d^1$ and Os$^{7+}$ 5$d^1$ ions, for instance, can be found in double-perovskite $fcc$ settings[26], but that is not the case for Ti$^{3+}$ or V$^{4+}$ 3$d^1$.

Here we individualize the group-5 lacunar spinels as a unique platform that makes it possible to illustrate how correlations shape many-body wave functions across a given group of the $d$ block—3$d$ to 4$d$ and 5$d$, for the same kind of leading electron configuration and in the same crystallographic setting. In particular, by expressing the many-body $M_4$-tetrahedron wave function in terms of localized single-ion $t_{2g}$ orbitals, we show that strong correlations yield a weight of 85–90% for the $t_{2g}^1 t_{2g}^2 t_{2g}^2 t_{2g}^2$ (i.e., V$^{4+}$V$^{3+}$V$^{3+}$V$^{3+}$) configurations in the vanadates; ferromagnetic double exchange occurs in this setting and yields near degeneracy of the low-lying low- and high-spin states—the $S = 1/2$ doublet is obtained as single-tetrahedron ground-state term only through a more advanced many-body treatment. In contrast, smaller $U/t$ ratios in the 4$d$ (Nb) and 5$d$ (Ta) systems take us away from the regime of strongly correlated electrons: in localized-orbital basis, we see that stronger charge fluctuations reduce the weight of $t_{2g}^1 t_{2g}^2 t_{2g}^2 t_{2g}^2$ resonances to ~ 50%; in molecular-orbital representation, significantly larger hoppings ($t$) and consequently larger bonding-antibonding splittings make that the *Aufbau* principle is to first approximation usable, with weights in the range of 65–75% for the $a_1^2 e^4 t_2^1$ configuration.

This result suggests a physical picture for the nonmagnetic states found in the Nb- and Ta-based lacunar spinels: as the $M_4$-cluster electrons are prone to stronger fluctuations with larger spatial spread, inter-cluster couplings are able to create "valence bond" spin singlets, as concluded from experiment[27–29]. The peculiar pseudospin structure must play a role in the superconductivity mechanism under pressure, which is speculated to be unconventional owing to closeness of magnetic states and spin fluctuations.

While it is clear that the on-site correlations affect the magnetic properties, they are only indirectly discernible in available experimental data, as already pointed out for susceptibility[21,22] and inelastic neutron scattering measurements[23] in in the previous section. More direct experimental insight into the details of the single-tetrahedron correlated electronic structure might be derived from resonant inelastic x-ray scattering experiments, as in the case of other clustered compounds, either $d$-[30] or $p$-electron[31–33] based.

Overall weights of ≳50% for the $t_{2g}^1 t_{2g}^2 t_{2g}^2 t_{2g}^2$ resonant valence structures suggest the $t$-$U$-$V$ model (or $U$-$tt'$-$VV'$ variants, where the primes denote inter-tetrahedral hopping matrix elements and Coulomb repulsion integrals) as means to explore correlation-induced symmetry breaking. Such numerical investigations could provide qualitative insights into the different types of low-temperature lattice symmetries realized in lacunar spinels and also into the polar properties of these materials. An analysis in terms of only three

molecular-like $t_2$ orbitals and one electron as in e.g., ref. 34 does not seem promising: according to our data, especially in the vanadates, the $(a_1^2 e^4) t_2^1$ description is not appropriate—the $M_4$ tetrahedron $t_2$ electron cannot be separated from the other six $d$-ion valence electrons, symmetry breaking should be rather described in terms of resonating holes in localized-orbital basis (i.e., $V^{4+}$ 'holes' in $V^{3+}$ 'background').

In sum, our quantum chemical data provide unparalleled specifics as concerns the correlated electronic structure of the group-5 lacunar spinels, well beyond the featureless $a_1^2 e^4 t_2^1$ picture circulated so far in the literature. Stronger correlations in the vanadates imply substantially less weight for the $a_1^2 e^4 t_2^1$ configuration as compared to the Nb and Ta compounds and render the molecular-orbital picture[8,9,35] inappropriate. Remarkably, spin-orbit coupling is still effective, even for the vanadates—the predicted fine structure with a splitting of $\approx 10$ meV between the $j \approx 3/2$ and $j \approx 1/2$ terms should be detectable experimentally. The stronger intersite fluctuations ($M^{3+} M^{3+} \rightarrow M^{2+} M^{4+}$) and the larger weight (65–75%) of the $a_1^2 e^4 t_2^1$ molecular-orbital configuration in the Nb and Ta spinels indicate that the $4d$ and $5d$ systems are closer to the Hartree-Fock limit. The different nature of valence-space charge fluctuations across the group-5 family of lacunar spinels should be relevant to inter-tetramer exchange; on-site charge fluctuations, for example, were shown to strongly renormalize intersite exchange in cuprate superconductors[36]. Assessing cooperative effects in group-5 lacunar spinels through the calculation of inter-cluster couplings will require approaches able to incorporate the correlated nature of the $M_4$-cluster ground states.

## Methods

The basic building block in the lacunar-spinel structure was described by a $[M_4 X_{28} Ga_6]^{25-}$ embedded cluster model ($M$ = V, Nb, Ta; $X$ = S, Se) (see Fig. 1). Experimentally determined high-temperature lattice parameters were adopted, as reported by Stefancic et al.[22] for $GaV_4S_8$ and by Pocha et al. for $GaNb_4Se_8$ and $GaTa_4Se_8$[7]. The influence of the surrounding bulk atoms was modeled by a finite point charge field (PCF) generated through the EWALD program[37,38]. A buffer region of 60 capped effective core potentials (cECPs) was set up between the quantum cluster and PCF (indicated by the smaller atoms in Fig. 1) (for details, see Section I of the Supplementary Information and Supplementary Dataset 1).

As initial step in our study, quasi-restricted orbitals (QROs[39]) were generated from an unrestricted Kohn-Sham B3LYP calculation for a single-configuration $S = 5/2$ state with initial-guess orbitals from Hueckel theory. The Hueckel guess ensures that the QROs fulfill $T_d$ point-group symmetry. Subsequently, 12 $[M_4]^{13+}$ molecular orbitals around the HOMO-LUMO gap were identified from the QROs and used as a starting point for CASSCF[17] calculations. Major convergence problems as encountered in earlier quantum chemical studies[11] are circumvented in this way. The valence-space multiplet structure was derived from state averaged (SA) CASSCF optimizations with those 12 orbitals and seven valence electrons defining the active space (denoted in quantum chemistry as (7e,12o) CASSCF), consequently corrected for dynamical correlation by $N$-electron valence 2nd order perturbation theory (NEVPT2)[18]. Both methods were accelerated by the resolution of identity (RI[40]) and chain-of-spheres (COS[41]) approximations for Coulomb and exchange integrals with automatically generated auxiliary basis sets[42]. All calculations were done using the program package ORCA, v5.0.3[43].

## Data availability

The quantum chemical data (coordinates of quantum clusters and point charge fields, ORCA outputs, and magnetic susceptibility simulations) generated in this study have been deposited in the RADAR database under the https://doi.org/10.22000/1655.

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

## Acknowledgements

We thank I. Kézsmárki, P. Fulde, and R. C. Morrow for discussions and U. Nitzsche for technical assistance. This work was supported by the German Research Foundation (Deutsche Forschungsgemeinschaft, DFG), Project No. 437124857. P.B. acknowledges funding from the DFG, Project No. 441216021.

## Author contributions

T.P. carried out the quantum chemistry calculations with assistance from P. B., U.K.R., and L.H. All authors were involved in writing the manuscript. U.K.R. and L.H. planned the project.

## Funding

## Competing interests

The authors declare no competing interests.
