## [Peer Review File · Nature Communications]

Resonating holes vs molecular spin-orbit coupled states in group-5 lacunar spinelsREVIEWER COMMENTS

Reviewer #1 (Remarks to the Author):

The group 5 Lacunar spinels are fascinating compounds exhibiting many exotic phenomena including cycloidal helimagnetism, polar/ferroelectric order, orbital ordering, valence bond ground states, and potentially unconventional superconductivity depending on the transition metal. These phenomena arise from correlated units that are tetrahedral clusters of transition metal ions, but the microscopic picture of the electronic configuration on these clusters and how electrons delocalized on the clusters interact with their neighbors is unclear. For instance, how best to understand magnetic exchange interactions between neighboring clusters that gives rise to the cycloidal magnetism in V compounds, or what is an appropriate wavefunction for intercluster magnetic singlets that presumably form valence bond type order in Nb and Ta compounds and ultimately give rise to (possibly) unconventional superconductivity under pressure?

Given the wide range of phenomena present in these materials, and the potential tunability that arises from the correlated molecular orbital units, it is essential to develop a consistent microscopic picture of the electronic states on those correlated units. Petersen and co-authors employ state-of-the-art quantum chemistry computations to investigate the electronic structure of group-5 transition metal tetrahedral clusters (M₄ clusters) and find that they are much more complex than the previously proposed single configuration state. In particular, multi-orbital configurations are essential, mixing in a significant proportion of a high spin excited state configurations with the low spin ground state. This effect is most pronounced in the V compounds, while a molecular orbital picture is more appropriate for Nb and Ta compounds. These findings should help to clarify an appropriate microscopic picture for the lacunar spinels and form the basis from which the intriguing correlated phenomena can be understood. Thus, this work could stand as an important reference point. The computational methods are state-of-the-art and appear to be of high quality. However, the paper is lacking in any direct connections or validations against the many published experiments on group 5 lacunar spinels.

It would be helpful to see some more direct comparisons of the results to experiment and an accompanying explanation of the physical interpretation of the findings here in terms of the magnetism. For instance, what are the implications of the findings here for interpretations of magnetic susceptibilities in these materials using a Curie-Weiss law? If there is significant mixing of low and high-spin states in the V compounds, this should affect that analysis. Can the authors provide an explanation for the high temperature deviation from Curie Weiss behavior reported for GaV₄S₈ [Philosophical Mag. 97 3428 (2017)]? What is the implication for the multi-configurational ground states for interpreting the reported Jahn-Teller transitions/orbital orderings in these compounds? Is the conclusion that the electronic state for V compounds contains a majority weight of $t_{2g}^1t_{2g}^2t_{2g}^2t_{2g}^2$ in the localized orbital representation consistent with a magnetic moment density on the V₄ tetrahedron that would give rise to the magnetic form factor found for GaV₄S₈ in PRB 102, 014410 (2020)?

There is by now also a wealth of spectroscopic characterization of the lacunar spinels that has been published and could be directly compared with energy scales from this work. Can the authors compare their results with published inelastic neutron scattering on GaV₄S₈ [PRB 104, 224425 (2021)] that may provide constraints on the lowest lying dipole allowed inter-orbital excitations? The calculated excitation energies for GaTa₄Se₈ could also be checked against published resonant inelastic x-ray scattering data. For instance the $j_{3/2} - j_{1/2}$ splitting reported from RIXS measurements is approximately 600 meV [Nat Commun. 8, 782 (2017)] and appears much larger than reported here, can the authors comment on this?

If the authors can address these issues by making more direct comparisons of their results with published experimental data and providing more explanation of the physical interpretation of the consequences of the multi-configurational electronic state for the magnetism in these lacunar spinels, then I believe it would make a much stronger case for the broad readership of Nature Communications and I would be happy to review this paper again.

Reviewer #2 (Remarks to the Author):

The manuscript reports on an impressively comprehensive study of the low energy eigenstates of group-5 lacunar spinels which shows very significant deviations from the generic $(t_2)^1$ configuration widely used in the literature, especially for the 3d electrons. While in some ways this is not a huge surprise (it is well expected that correlations are stronger in the 3d than in the 4d and 5d shell), the interesting aspect to me is how big the effect is.

The work seems to be well implemented and carried out and I believe it will have a significant effect on the modelling of these materials in future work. I recommend that the paper be accepted.

Reviewer #3 (Remarks to the Author):

The manuscript tries to tease out trends in correlations and wave functions when going from 3d to 5d elements by focusing on materials that are otherwise as similar as possible. They perform state-of-the-

art quantum chemistry calculations for group-5 lacunar spinels and analyze the resulting ground state in terms of orbitals localized at single transition-metal atoms vs. shared by a tetrahedron of four ions.

The most important results are:

- i) A less correlated picture obtained by filling shared orbitals is more appropriate to the 4d and 5d cases than to 3d, i.e. vanadium.
- ii) Spin-orbit coupling is more important for later elements, but still noticeable for V.

Technically, the paper is very sound. They use state-of-the-art quantum-chemistry methods and the group are expert on them. They also adapt the method to the compound studied, so, results can be expected to hold.

The authors try to put their findings into context and -- plausibly enough -- argue that the differences in the wave

functions will affect material properties or results from models based on them.

I find the paper solid, helpful and interesting. However, it does not seem very exciting. The trends on correlations/spin-orbit-coupling with respect to position in the periodic table are exactly as one would have expected. It is of course valuable to have them quantified (e.g. how much SOC remains relevant for V, or how much of the wave function continues to be due to localized orbitals for heavier elements) and for this purpose, it is a good

idea to choose a material system where they are all comparable. Nevertheless, it is not clear to me how much interest this will draw.

In order to better link the numerical results to the

big general-interest picture, it would be very nice to have specific examples, which model calculation(s) or experiment(s) will have to be reassessed in view of the present results.

Small things that I would have found helpful when reading the manuscript:

- Explain in Sec. II what can be inferred from the need of more advanced techniques for V.
-) Try to find another way to express "(...)" single-configuration" in the abstract, because a more general reader would not understand it easily this early in the paper.

-) In the second paragraph of Sec. I, clarify what is what in "either localized, atomic-like or multisite orbitals", because "localized" could mean almost anything here.

Reviewer #4 (Remarks to the Author):

It was rather difficult for me to determine until reading well into the manuscript that in this theoretical work the authors study the valence structure of magnetic ions using a multiconfigurational complete active space self-consistent field (CASSCF) method with an active space of seven electrons in twelve orbitals (7e,12o)-CAS. (It was not easy to find this! It took until page 5!) Then, shortly afterwards, the authors state "accounting for dynamical

correlation effects in the scheme of post-CASSCF N -electron valence perturbation theory (NEVPT2) corrects this state ordering." There is no description or reference to supplemental information for details around the text of these not-so-well known approaches.

Comments: The title is unclear as "Luxuriant" means "rich and profuse in growth", but I don't see any biology or anything growing in a spinel. As a researcher working in the field, I have no idea what this paper is about from the title, other than spinels are involved and some of the orbital details are investigated.

The abstract also has many English usage problems and is exceptionally unclear. For example, "valence structure of magnetic centers" is not at all common language in the field. The following sentence "It may pertain to" is also not well written—"pertain to" is quite vague. "explore the inner structure of magnetic moments" sounds like the authors are trying to write an astrophysical paper about stars or a nuclear physics paper about the nucleus. How about "orbital structure of magnetic moments"? That is much clearer. Another issue is "...M_4 units...". This notation is not introduced or known to the wider audience that might read this paper. I would suggest the authors rewrite the entire abstract so a general reader can understand what was done (methods used, main results—concretely—not "shape vibronic...."), create a new title that captures the essence of the study and does not use words such as "luxuriant" which is inappropriately applied.

Then, in the first paragraph, "desiderata" is used. A great word for an English professor to use in class, but not for the introduction to a physics paper. I think the writing of this paper can be summarized as saying many "fancy" words are used inappropriately, while the overall scientific points/messages are

unacceptably vague for me. I should not have to work so hard and stretch to try to find the science in the paper.

I would suggest the authors follow a more standard format to the intro to their paper: (1) Summarize relevant literature and identify “gaps” or “open issues”. (2) Explain how the work described in the manuscript will fill the gap or settle the issue. (3) Go on to provide the technical details and main results of the work. (4) Summarize, and state the important implications for the field and broader readers.

I cannot recommend the manuscript for publication without essentially a complete re-write.

— Response to the report of Reviewer #1 —

We are pleased with the positive comments made by Reviewer #1, e. g., that our “findings should help to clarify an appropriate microscopic picture for the lacunar spinels and form the basis from which the intriguing correlated phenomena can be understood”. Her/his few suggestions and the extensions that we made in our analysis/discussion in response to those suggestions are addressed in the following.

“... .. However, the paper is lacking in any direct connections or validations against the many published experiments on group 5 lacunar spinels.”

The many specific suggestions regarding the implications of our results are answered point by point in the following. But we should first state that our *ab initio* calculations and the insights derived from there clearly constitute the first major step to understand the correlated electronic structure of such tetrahedral-cluster compounds and in-depth further analysis will still require separate demanding computational studies and possibly methodological advances.

We agree with the implicit critique by the reviewer and believe that next steps in tackling the electronic structures and physical properties of the lacunar spinels will have to address all those issues in detail. But we also feel that it is more important to inform the interested community that the electronic-structure picture and conceptual frame used so far for these compounds are qualitatively incorrect and therefore seriously deficient than explaining in great detail various experimental features.

We have improved the structure of our manuscript in many places and discuss now a number of implications of our result for the interpretation of experimental data in answer to the suggestions made by Reviewer #1. We thank Reviewer #1 for helping us to better structure our thoughts and our representation of the electronic structures of these materials.

“It would be helpful to see some more direct comparisons of the results to experiment and an accompanying explanation of the physical interpretation of the findings here in terms of the magnetism. For instance, what are the implications of the findings here for interpretations of magnetic susceptibilities in these materials using a Curie-Weiss law? If there is significant mixing of low and high-spin states in the V compounds, this should affect that analysis. Can the authors provide an explanation for the high temperature deviation from Curie Weiss behavior reported for GaV₄S₈ [Philosophical Mag. 97 3428 (2017)]?”

Indeed, the deviation from Curie-Weiss behavior seen in the experiments mentioned by Reviewer #1 may be attributed to populating excited-state levels of the V₄ clusters. As the reviewer correctly observes, Fig. 2 of our manuscript indicates a denser set of low-lying excited states in GaV₄S₈ as compared to GaNb₄Se₈ and GaTa₄Se₈. Using the computed level splittings, it is possible to derive susceptibility curves, as shown for example in the plot included hereby. Significant deviations from Curie-Weiss behavior are seen as temperature increases. However, quantitatively, our curve under-estimates the effect observed experimentally. We attribute this underestimation to:

- (i) correlation effects not fully accounted for in the CASSCF+NEVPT2 correlation treatment; even more sophisticated calculations based on even larger orbital active spaces in CASSCF or/and post-CASSCF configuration interaction are, however, extremely demanding computationally and would represent a separate project in our view;
- (ii) vibronic effects that are completely neglected in our present study.

In the revised manuscript, a few phrases on the implications of the rather dense set of low-lying excited states in the vanadium compounds are included in the very end of section II.A (lines 111-115) and in the supplemental material.

Figure 1: Calculated magnetic susceptibility curve for GaV_4S_8 .

“What is the implication for the multi-configurational ground states for interpreting the reported Jahn-Teller transitions/orbital orderings in these compounds? Is the conclusion that the electronic state for V compounds contains a majority weight of $t_{2g}^1t_{2g}^2t_{2g}^2t_{2g}^2$ in the localized orbital representation consistent with a magnetic moment density on the V_4 tetrahedron that would give rise to the magnetic form factor found for GaV_4S_8 in PRB 102, 014410 (2020)?”

Jahn-Teller couplings, the low-temperature electronic structure(s), and the magnetic form factors represent aspects well beyond the purpose of our present study. All that will require additional extensive investigations and much more expensive computations. But one important statement that can be made in light of numerical quantum chemical data that we have done so far is the following: for the low-temperature vanadate electronic structure(s), we need to establish if/how the 'resonating hole' gets localized (i. e., the $t_{2g}^1t_{2g}^2t_{2g}^2t_{2g}^2$ hole in localized-orbital basis), and not the t_2^1 electron.

In other words, the $(a^2e^4)t_2^1$ molecular-orbital picture is not appropriate in vanadates, the t_2^1 electron discussed for example in the paper mentioned by Reviewer #1 [PRB 102 014410 (2020)] cannot be taken alone, separated from the other 6 d -ion valence electrons. Our feeling is simply that our colleagues do not look for the right thing in their experimental data; the DFT-based $(a^2e^4)t_2^1$ molecular-orbital picture only leads them astray.

A few lines pointing out such aspects and implications are now added in the end of the 3rd paragraph in “Discussion” (lines 207-212).

“There is by now also a wealth of spectroscopic characterization of the lacunar spinels that has been published and could be directly compared with energy scales from this work. Can the authors compare their results with published inelastic neutron scattering on GaV_4S_8 [PRB 104, 224425 (2021)] that may provide constraints on the lowest lying dipole allowed inter-orbital excitations?”

Direct comparison between our present results and the INS data by Pokharel *et al.* is not possible as the INS measurements probe spin wave excitations to deduce the magnetic couplings between adjacent V_4 clusters. In our present study, only a single V_4 unit is explicitly considered — inter-tetrahedral couplings are out of reach for this material model. We have addressed this issue in the “Discussion”, lines 196-201.

“The calculated excitation energies for GaTa_4Se_8 could also be checked against published resonant inelastic x-ray scattering data.”

In the study by Jeong *et al.* mentioned by Reviewer #1, RIXS measurements and accompanying Hubbard-model calculations were performed. Their assignment of the most prominent RIXS spectral features does not really agree with what our computations indicate. This is now pointed out in the second to last paragraph in section II.A (lines 105-108).

— Response to the report of Reviewer #2 —

“The manuscript reports on an impressively comprehensive study of the low energy eigenstates of group-5 lacunar spinels which shows very significant deviations from the generic (...) t_2^1 configuration widely used in the literature, especially for the 3d electrons. While in some ways this is not a huge surprise (it is well expected that correlations are stronger in the 3d than in the 4d and 5d shell), the interesting aspect to me is how big the effect is. The work seems to be well implemented and carried out and I believe it will have a significant effect on the modelling of these materials in future work. I recommend that the paper be accepted.”

Indeed, the effect is “big”: stronger correlations in the (3d) vanadates make that the molecular-orbital, single-configuration t_2^1 picture presently circulated in the literature breaks down completely. In other words, our work redefines the conceptual frame within which the properties of the vanadium compounds should be addressed.

— Response to the report of Reviewer #3 —

“The manuscript tries to tease out trends in correlations and wave functions when going from 3d to 5d elements by focusing on materials that are otherwise as similar as possible. They perform state-of-the-art quantum chemistry calculations for group-5 lacunar spinels and analyze the resulting ground state in terms of orbitals localized at single transition-metal atoms vs. shared by a tetrahedron of four ions.

The most important results are:

i) A less correlated picture obtained by filling shared orbitals is more appropriate to the 4d and 5d cases than to 3d, i.e. vanadium.

ii) Spin-orbit coupling is more important for later elements, but still noticeable for V.

Technically, the paper is very sound. They use state-of-the-art quantum-chemistry methods and the group are expert on them. They also adapt the method to the compound studied, so, results can be expected to hold.

The authors try to put their findings into context and -- plausibly enough -- argue that the differences in the wave functions will affect material properties or results from models based on them.

I find the paper solid, helpful and interesting.”

We are pleased with the positive comments made by Reviewer #3. Indeed, crystallographically and ‘band’-filling-wise, the group-5 lacunar spinels are “as similar as possible”. Yet, we show that the strength of *d*-electron correlations brings crucial ‘dissimilarity’ across the series.

“However, it does not seem very exciting. The trends on correlations/spin-orbit-coupling with respect to position in the periodic table are exactly as one would have expected. It is of course valuable to have them quantified (e.g. how much SOC remains relevant for V, or how much of the wave function continues to be due to localized orbitals for heavier elements) and for this purpose, it is a good idea to choose a material system where they are all comparable. Nevertheless, it is not clear to me how much interest this will draw.”

Indeed, we get things “quantified”. And point out in this way qualitatively different regimes: strongly correlated resonating valence structures for 3d ions vs ‘dressed’ (i. e., weakly to moderately correlated) $j=3/2$ spin-orbital states in delocalized, molecular-orbital basis for 4d and 5d species.

That others “would have expected” already “exactly” what our analysis reveals and quantifies is not impossible. Still, we are not aware of any other group discussing, quantifying, and reporting the kind of physics we are describing in this manuscript.

“In order to better link the numerical results to the big general-interest picture, it would be very nice to have specific examples, which model calculation(s) or experiment(s) will have to be reassessed in view of the present results.”

For better perspective on implications deriving from our *ab initio* results, we point out now in the last part of the 3rd paragraph in “Discussion” (lines 207-212) that as concerns symmetry breaking and charge localization at low temperature, what we should search for in the experimental data is how the ‘resonating hole’ in localized-orbital basis gets localized, not the t_2^{-1} electron in molecular-orbital representation; the DFT-based (a^2e^4) t_2^{-1} single-configuration molecular-orbital representation only leads researchers astray.

Additionally, in the last paragraph in section II.A, (lines 99-116) we make connection between the *ab initio* findings as concerns the single-tetrahedron multiplet structure and RIXS plus magnetic-susceptibility measurements. The need for further RIXS measurements as well as possible implications for INS data of GaV₄S₈ are additionally explained in the “Discussion” (lines 196-201).

“Small things that I would have found helpful when reading the manuscript:

- Explain in Sec. II what can be inferred from the need of more advanced techniques for V.

-) Try to find another way to express “(...)t₂¹ single-configuration” in the abstract, because a more general reader would not understand it easily this early in the paper.

-) In the second paragraph of Sec. I, clarify what is what in “either localized, atomic-like or multisite orbitals”, because “localized” could mean almost anything here.”

We thank Reviewer #3 for these last three suggestions.

The first seems to be closely related to the previous question. That is now better detailed in the 3rd paragraph in "Discussion" (implications as concerns the low-temperature charge distribution) and in the last paragraph in section II.A (implications as concerns the interpretation of excited-state level structures).

As concerns the use of the t_2^1 notation in our abstract, that part is modified now in the revised version of the manuscript, from "...richer than the generic t_2^1 single-configuration description..." to "...richer than the single-electron, single-configuration description...".

Finally, "localized, atomic-like" has been changed into "localized, site-centered" (line 34).

— Response to the report of Reviewer #4 —

"It was rather difficult for me to determine until reading well into the manuscript that in this theoretical work the authors study the valence structure of magnetic ions using a multiconfigurational complete active space self-consistent field (CASSCF) method with an active space of seven electrons in twelve orbitals (7e,12o)-CAS. (It was not easy to find this! It took until page 5!) Then, shortly afterwards, the authors state "accounting for dynamical correlation effects in the scheme of post-CASSCF N-electron valence perturbation theory (NEVPT2) corrects this state ordering." There is no description or reference to supplemental information for details around the text of these not-so-well known approaches."

These omissions are now incorporated into the revised manuscript, we thank the reviewer for pointing out those lacunae:

- that quantum chemical methods are used is explicitly mentioned now in the abstract and introduction (line 32)
- that multiconfigurational wave-functions are analyzed is mentioned already in introduction (line 33)
- references to the general CASSCF and NEVPT2 methods were added in the beginning of section II.A (lines 83 and 88) as well as further explanations in the supplemental material

"The title is unclear as "Luxuriant" means "rich and profuse in growth", but I don't see any biology or anything growing in a spinel. As a researcher working in the field, I have no idea what this paper is about from the title, other than spinels are involved and some of the orbital details are investigated.

The abstract also has many English usage problems and is exceptionally unclear. For example, "valence structure of magnetic centers" is not at all common language in the field. The following sentence "It may pertain to" is also not well written — "pertain to" is quite vague. "explore the inner structure of magnetic moments" sounds like the authors are trying to write an astrophysical paper about stars or a nuclear physics paper about the nucleus. How about "orbital structure of magnetic moments"? That is much clearer. Another issue is "...M_4 units...". This notation is not introduced or known to the wider audience that might read this paper. I would suggest the authors rewrite the entire abstract so a general reader can understand what was done (methods used, main results—concretely—not "shape vibronic..."), create a new title that captures the essence of the study and does not use words such as "luxuriant" which is inappropriately applied.

Then, in the first paragraph, "desiderata" is used. A great word for an English professor to use in class, but not for the introduction to a physics paper. I think the writing of this paper can be summarized as saying many "fancy" words are used inappropriately, while the overall scientific points/messages are unacceptably vague for me. I should not have to work so hard and stretch to try to find the science in the paper.

I would suggest the authors follow a more standard format to the intro to their paper: (1) Summarize relevant literature and identify "gaps" or "open issues". (2) Explain how the work described in the manuscript will fill the gap or settle the issue. (3) Go on to provide the technical details and main results of the work. (4) Summarize, and state the important implications for the field and broader readers.

I cannot recommend the manuscript for publication without essentially a complete re-write."

We thank Reviewer #4 for pointing out potential problems as concerns some of the terms ("luxuriant", "desiderata" etc.) used in Abstract and Introduction. It all has to do with being fed too much Latin and French in secondary school and/or high-school — besides not really enjoying those classes, it brings us just trouble nowadays.

Regarding the title, we removed "luxuriant" and propose instead "Protean nature of electronic correlations: resonating holes vs molecular spin-orbital states in group-5 lacunar spinels". As we are reporting novel and uncommon observations, there is a need to signal that. Employing our otherwise obsolete classical education we now realize that the diversity of electronic states could nicely be depicted by reference to the aquatic god capable of assuming many different forms, sometimes appearing on remote islands. We hope this idea is better received than the "luxuriant" word.

In Abstract, we made the following modifications :

- “the valence structure of magnetic centers” → “the valence electronic structure of magnetic centers”,
- “It may pertain to” → “It may refer to”,
- “M₄ units” → “M₄ tetrahedral clusters” (the presence of “multisite magnetic units in the form of tetrahedral tetramers” is mentioned a couple of lines above, already in the older version of the manuscript).

As concerns the suggestion of replacing “the inner structure of magnetic moments” by “orbital structure of magnetic moments”: our analysis refers not only to multiorbital effects but also to multisite and multiconfigurational physics. We would therefore prefer to keep the term “inner” in that phrase; “orbital” significantly narrows the perspective.

We have additionally rewritten significant parts of our introduction section. The word “desiderata” has been removed in this process (line 13). We trust Reviewer #4 will find this new version of our synopsis appropriate.

REVIEWERS' COMMENTS

Reviewer #1 (Remarks to the Author):

The authors have taken time to thoughtfully address and reply to all of my concerns. The revised manuscript is improved in its clarity and following my previous report I now recommend publication in Nature Communications.

Reviewer #3 (Remarks to the Author):

I am afraid my report is not going to be very helpful, as I can't add much beyond my previous one. I continue to find the paper's

careful quantitative analysis impressive, helpful and valuable, but it is still not clear to me how exciting it is.

The authors are (understandably enough) miffed that I suggested that the basic trends are "as expected". Let me clarify: It has, for example, been taught for decades that L-S-coupling

(dominated by Hund's rule and thus correlations) goes over into j-j coupling (dominant spin-orbit coupling) with heavier elements. So, the qualitative trend observed in the paper corresponds certainly to expectations, even if there is at this time no longer "any other group discussing" this. The main new findings of the present paper are thus certainly the **quantitative** ones, e.g. how big the impact of correlations is for V and how much spin-orbit coupling survives.

Referee 2, for instance, appears to have been impressed with the impact of correlations on V . (That I was less surprised may have to do with my scientific background, so it may be me.) I had hoped that the new version would bring the importance of the quantitative findings more clearly than it does.

BTW, I do not think "protean" is a good idea: (a) it is likely not to be too widely understood and (b) to me, it carries some connotation of fickleness (as if the correlations did sometimes this and sometimes that), which goes exactly against the main strength of the paper: a deep understanding when they do what exactly.

Reviewer #4 (Remarks to the Author):

I have appreciate the authors' effort to improve the readability of the manuscript and their response to my comments and the other referees. Now that I am able to understand the study and modifications to the manuscript based on the other three referees' comments, I have a favorable opinion of the work.

I think the study based on quantum chemistry methods provides a viewpoint complementary to what is commonly adapted in the field and example of the vanadium case illustrates that for 3d elements the differences can be qualitatively significant. (See Fig. 2(a) and Fig 3, for example.) The modified discussion section helps to put these differences in context and draw out the broader implications for the readership of nature communications. I am happy to recommend publication of the work.

— Response to the report of Reviewer #1 —

“The authors have taken time to thoughtfully address and reply to all of my concerns. The revised manuscript is improved in its clarity and following my previous report I now recommend publication in Nature Communications.”

We are pleased with the positive comments from Reviewer #1 and would like to thank him/her once more for his/her valuable comments.

— Response to the report of Reviewer #3 —

“I am afraid my report is not going to be very helpful, as I can't add much beyond my previous one. I continue to find the paper's careful quantitative analysis impressive, helpful and valuable, but it is still not clear to me how exciting it is.

*The authors are (understandably enough) miffed that I suggested that the basic trends are "as expected". Let me clarify: It has, for example, been taught for decades that L-S-coupling (dominated by Hund's rule and thus correlations) goes over into j-j coupling (dominant spin-orbit coupling) with heavier elements. So, the qualitative trend observed in the paper corresponds certainly to expectations, even if there is at this time no longer "any other group discussing" this. The main new findings of the present paper are thus certainly the *quantitative* ones, e.g. how big the impact of correlations is for V and how much spin-orbit coupling survives.*

Referee 2, for instance, appears to have been impressed with the impact of correlations on V. (That I was less surprised may have to do with my scientific background, so it may be me.) I had hoped that the new version would bring the importance of the quantitative findings more clearly than it does.”

As the reviewer notes, his/her concerns were already raised in the 1st round of referee reports. Besides discussing the differences between “resonating holes” and a “molecular-orbital picture” in the “Discussion” section, we also added a visual impression of both situations in the inset of Figure 1. Beyond that, we feel there is nothing we can add to our reply from the 1st round at this point.

“BTW, I do not think "protean" is a good idea: (a) it is likely not to be too widely understood and (b) to me, it carries some connotation of fickleness (as if the correlations did sometimes this and sometimes that), which goes exactly against the main strength of the paper: a deep understanding when they do what exactly.”

We understand the remark of reviewer #2 concerning the title of the manuscript. We therefore use a modified title in the revised version submitted (the new title is shorter and does not contain the word “protean”).

— Response to the report of Reviewer #4 —

“I have appreciate the authors' effort to improve the readability of the manuscript and their response to my comments and the other referees. Now that I am able to understand the study and modifications to the manuscript based on the other three referees' comments, I have a favorable opinion of the work.

I think the study based on quantum chemistry methods provides a viewpoint complementary to what is commonly adapted in the field and example of the vanadium case illustrates that for 3d elements the differences can be qualitatively significant. (See Fig. 2(a) and Fig 3, for example.)

The modified discussion section helps to put these differences in context and draw out the broader implications for the readership of nature communications. I am happy to recommend publication of the work.”

We are pleased with the positive comments from Reviewer #4 and would like to thank him/her for recommending our manuscript for publication.